# Thinning Levels of Laurel Natural Regeneration to Establish Traditional Agroforestry Systems, Ecuadorian Amazon Upper Basin



Álvaro Cañadas-López [1,*], Paul Gamboa-Trujillo [1], Santiago Buitrón-Garrido [1], Byron Medina-Torres [1], J. Jesús Vargas-Hernández [2] and Christian Wehenkel [3]

1   Carrera en Ingeniería en Recursos Naturales, Universidad Central del Ecuador, Quito EC170136, Ecuador; jpgamboa@uce.edu.ec (P.G.-T.)
2   Colegio de Posgraduados, Posgrado en Ciencias Forestales, Texcoco 56230, Mexico
3   Instituto de Silvicultura e Industria de la Madera, Universidad de Juárez del Estado de Durango, Durango 34120, Mexico
*   Correspondence: agcanadas@uce.edu.ec; Tel.: +593-93-906-8644

**Abstract:** (1) Background: The *Cordia alliodora* (Ruiz & Pav.) Oken (laurel) natural regeneration management is a widespread practice among smallholders in the Amazon upper basin for the establishment of traditional agroforestry systems. This tree management approach is opposite to the development project proposals that contemplate reforestation with nursery seedlings in the Amazon region. The present study evaluated the effects of thinning levels on the diameter and basal area increment of laurel in a traditional agroforestry system; (2) Methods: A randomized complete block design with three replications was used, and the target variables were the growth rate of diameter at breast height (DBH) and the basal area of trees. Twelve square field plots, 400 m$^2$ each, were established in a plot network covering a 1.0 km × 0.5 km area. Three thinning levels were applied: light, moderate, and intensive thinning; (3) Results: The traditional agroforestry system investigated was characterized by a marginal growth of laurel trees, with an average yield of 125.26 m$^3$ ha$^{-1}$ (±15.39) and MAI of 13.92 m$^3$ ha$^{-1}$ at 9 years of age with a tree density of 418 trees ha$^{-1}$. The average value of the relation between the number of trees and the basal area removed (NG value) was 1.15, with small variation among plots, so all of them were thinned from below. The intensive thinning treatment, leaving 200 trees ha$^{-1}$, caused the greatest annual increase in individual tree DBH (2.03 cm) and basal area (61.37 cm$^2$) in both absolute and relative terms and improved tree height/diameter ratio; (4) Conclusions: Management of laurel natural regeneration for the establishment of traditional agroforestry systems could be improved by thinning at early ages, leaving 200 well-spaced laurel trees per ha.

**Keywords:** *Cordia alliodora*; height/dimeter ratio; relative growth rate; stand density; thinning intensities

## 1. Introduction

The net deforestation rate in Ecuador has been high, with an annual rate of 1.5% for the 1990–2000 period [1]. In the Amazon region, the forces that accelerated deforestation between 1995 and 1997 within the Sumaco Protected Forest of the canton Loreto were the increase in land use for agriculture and livestock in rich soils around rivers, flat terrains, indigenous communities, and, to a lesser extent, sloping lands. From 1997 to 2007, the deforestation rate fell, changing the deforestation forces [2]. For the 2014–2016 period, the deforestation rate in the Ecuadorian Amazon region decreased to 0.48% [3]. These forces molding the land-use change and land-forest-cover lost left extensive areas of secondary forest. The potential value of secondary forest regeneration, which covers extensive areas in the Ecuadorian Amazon Region, is not always recognized [4–7]. Nevertheless, neotropical

native species frequently grow on fallow, degraded land and could be used for economic and ecological rehabilitation of farmland to restore native forest ecosystems [8]. A well-managed fallow could be an effective reforestation method and could be used as an option for establishing agroforestry systems [9].

Forest succession is a fundamental ecological process, which has significant implications for sustainable natural resource management as well as for the biological, biophysical, and biogeochemical processes in an ecosystem [10]. Knowledge about the regeneration of a given forest is of great importance to develop strategies for forest management and biodiversity conservation [11]. The regeneration of forest species is the result of complex ecosystem processes involving reproduction and seed dispersal in a given environmental context; however, the interaction between ecological and social factors are particularly important at the landscape scale to forge natural regeneration [9]. Population density of saplings and trees are essential to determine the structure and regeneration stages of any forest community [12]. Amazon forests are known for their rapid recovery after the abandonment of pasture and agricultural areas; resprouting from shoots and roots, soil seed bank and seed dispersal are the main mechanisms responsible for restoration [13].

Cañadas [14] points out that the economic value of the secondary forests could be increased by establishing agroforestry systems and that the economic value of the fallow land could be improved by utilizing the natural regeneration of *Cordia alliodora* (Ruiz & Pav.) Oken, locally know by the vernacular name, laurel. Within forest gaps, laurel is a pioneer species, which, together with other species of all sizes, emerges from the gaps. Under neotropical forest conditions, laurel is limited to gap areas greater than 50 m$^2$ [15]. The laurel under secondary forest conditions is characterized by the ability to assume different growth patterns, having an immense resilience [16]. In crop fields, the laurel seedlings remain dormant under the crop shadow. After harvesting the crop, laurel reacts immediately to the increase in sunlight and grows with renewed vigour to take over the canopy layer [17]. Amazon smallholders know diverse management techniques to promote tree growth; one of them is the natural regeneration of secondary tree species with economic value. Likewise, in Costa Rican farms, laurel from natural regeneration can be seen commonly in several agroforestry systems, such as pasture lands, in small coffee and cocoa production, and in sugar cane crops [18]. Villa et al. [19] found that density management of the laurel tree components in the traditional agroforestry systems is handled by the native farmers fitting a specific crop system, but no silvicultural techniques are used to improve tree form or vigour. Hummel [20] suggested that laurel natural regeneration systems leading to even-age stands are good candidates for conducting thinning studies using techniques applied to temperate forests because laurel trees show a decrease in diameter growth with increasing tree densities. Somarriba et al. [21] emphasized the need to do more systematic studies analyzing population dynamics, including seedling and tree survival, as well as thinning and harvest methods for laurel in agroforestry systems.

Little is known about the ecologic and economic potential of laurel natural regeneration in the Amazon upper basin. This approach is opposed to the predominant schemes imposed by development projects that contemplate reforestation with nursery seedlings. Surprisingly, these forestry projects had little sustainable effect on the local situation in the Amazon region [6]. Tree density thinning weights provide data which can be used to predict the effects of different regenerated tree densities and thinning designs on laurel stem growth to improve agroforestry systems. This current study aims to evaluate the effects of thinning levels on stem diameter and basal area growth of laurel within traditional agroforestry systems. The results should provide information for the establishment and better management of traditional agroforestry systems within the Ecuadorian Amazon region as an alternative to the classic reforestation projects.

## 2. Materials and Methods

### 2.1. Location and Conditions of the Experimental Area

This study was conducted in the province of Napo, parish of Wamaní, Guagua Sumaco and Pacto Sumaco (Latitude 0°42′08″ S and Longitude 77°44′04″ W), at 1160 m above sea level (Figure 1).

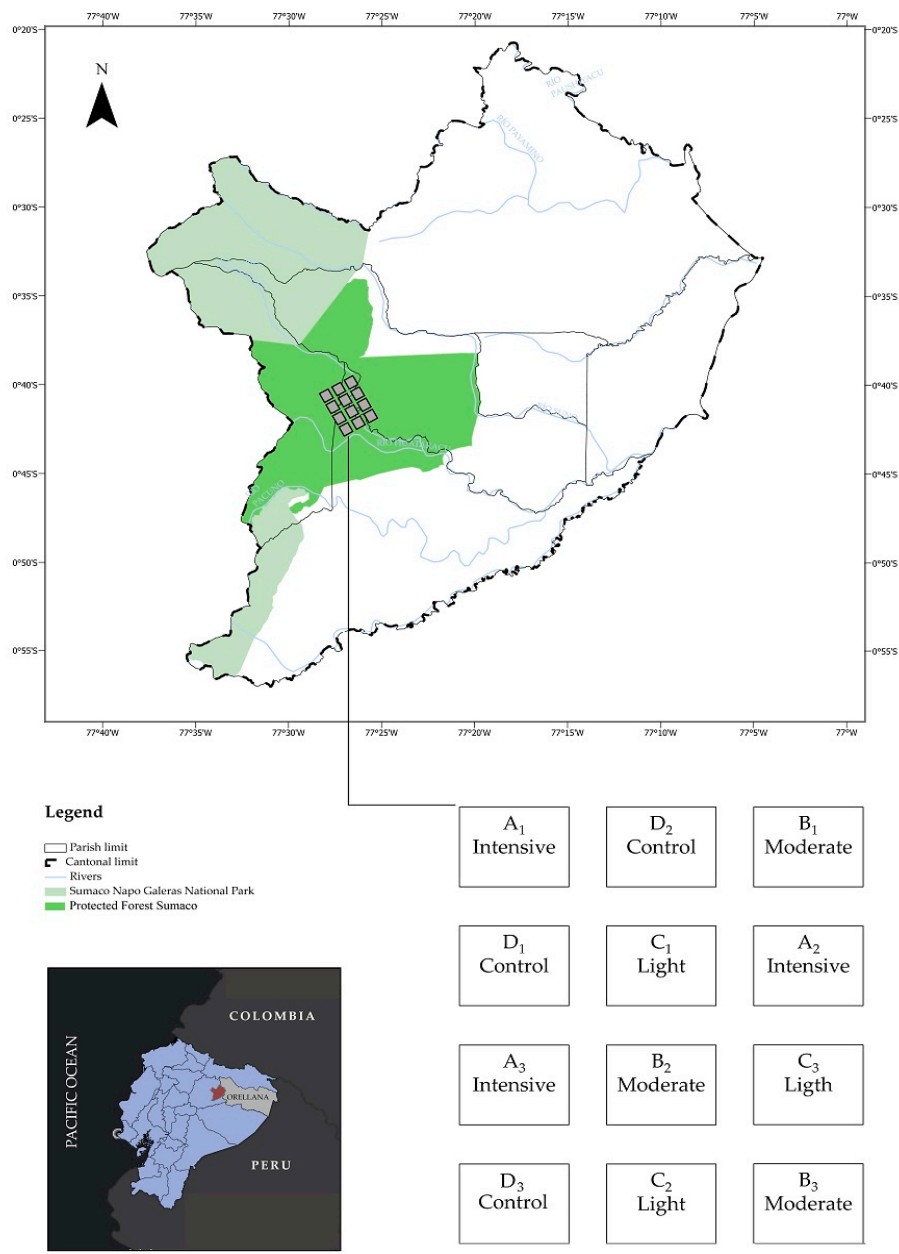

**Figure 1.** Location and design of the thinning trial for laurel, including areas with different management status in canton Loreto, Orellana Province.

The average annual temperature is 21 °C, and the mean precipitation is 4000 mm per year, which is evenly distributed over the year. The soils in the investigation area are quite homogenous and can be classified as Hydrandepts [22]. A high release of aluminum and hydrogen cations contributes to the acidification of the soils. Severe weathering and leaching processes can be observed due to the high precipitation rate and the high capacity of water retention in soil [14].

The traditional agroforestry system was originated by natural regeneration and traditional management of laurel. It consists of the cutting of one hectare of primary forest

with some laurel trees left to reinforce natural regeneration of this species. The research area was used for the cultivation of *Zea mays* L. (maize) the first year and *Solanum quitoense* Lam. (naranjilla) the next four years. In this agricultural period, natural regeneration and growth of laurel was allowed, forming a relatively homogenous forest and establishing the traditional agroforestry system [22].

## 2.2. Statistical Design

A randomized complete block design with three replications was used. The target variables were growth rate of diameter at breast height (DBH) and basal area. Figure 1 shows the spatial distribution of plots into the Sumaco Protected Forest (Figure 1). The plot network covered a 1.0 × 0.5 km (300 m northbound and 100 m east and westbound) land area. The plots' dispersion is explained by the search for the most homogeneous plots possible as an agroforestry system as described above. The twelve square field plots, 400 m$^2$ each with a net plot size of 324 m$^2$, were established within this area [23–26]. Three thinning levels were applied: control (D plots, no thinning was done); light thinning (C plots, low thinning affecting low-diameter trees); moderate thinning (B plots, restrained thinning affecting low- and medium-diameter trees); and intensive thinning (A plots, affecting some codominant and high-diameter trees). The plot distribution in each of the thinning levels plots was the centre of the identified agroforestry system.

## 2.3. Quantitative and Qualitative Thinning Indices

Quantitative thinning weights were calculated from the trees removed in relation to the original stand before thinning, expressed in number of trees (TN), basal area (TG), and volume (TV). The qualitative thinning weight can be described by the ratio between the relative number of trees removed and their relative basal area. This value is also described as the relationship between number of trees and basal area, *NG* [27]. The following formula was employed:

$$NG = \frac{N_{out}/N_{Total}}{G_{out}/G_{Total}} \tag{1}$$

where $N_{out}$ = number of trees removed, $Nt_{otal}$ = total number of trees in the original plot, $G_{out}$ = basal area of the removed stock, and $G_{total}$ = basal area of the total stock.

## 2.4. Information and Data Collection

All trees in the twelve plots were counted and identified with a number marked at breast height (1.3 m) with an inverted T to ensure that the same point was maintained to measure diameter at breast height (DBH, cm). Tree height (H, m) was estimated with a Haga hypsometer. Initial measurements and thinning were carried out in April of 2015. A second measurement was done in April of 2016, when the laurel trees were ten years old. Average height ($H_g$, m); average DBH ($D_g$, cm); average basal area (G, m$^2$); top DBH ($D_o$, cm); and top H ($H_o$, m) were identified for each plot. Mean annual increment [8] (MAI, m$^3$ ha$^{-1}$ year$^{-1}$) and $H_o/D_o$ ratios per diameter class were measured. Individual tree volume was calculated using basal area, height, and form factor (0.7) for laurel at the Ecuadorian Amazon upper basin.

## 2.5. Statistical Analyses

A variance analysis was performed with the General Lineal Models procedure, using SAS software (V 9.4, SAS Institute Inc., Cary, NC, USA), for the individual values of trees in each plot and a one-way ANOVA model with sub-sampling (trees within each plot) to determine if there are significant effects of thinning weights on diameter and basal area growth. Because differences in the initial diameter of residual trees were found between plots assigned to different thinning weights, a statistical analysis of relative growth rate in

diameter and basal area was also performed to exclude the effect of the initial size of laurel trees. The relative growth rate (RGR) was calculated with the following equation:

$$\text{RGR} = \frac{\left(lnD_f - lnD_i\right)}{T_f - T_i} \tag{2}$$

where $lnD_f$ and $lnD_i$ are the natural logarithm of the final and initial values, respectively, of the analyzed variable (DBH or basal area), and $T_f$ and $T_i$ are the corresponding dates of final and initial measurement. For variables in which a significant thinning effect was found, a Tuckey means comparison test was performed, with a $p = 0.05$ probability level.

To analyze the relationship of DBH and area basal increase with plot density in terms of number of trees (N) and basal area (G), Pearson's linear correlation (Proc Corr SAS) was calculated using absolute and relative plot values ($n = 12$) for DBH and basal area increase. In addition, regression models (linear and quadratic) for absolute and relative growth in DBH and basal area on N and G plot values were adjusted and compared (Proc Reg, SAS) in terms of the following indicators: Mallow's C(p), Akaike Information Criterium (AIC), Bayesian Information Criterium (BIS), and Schwarz Bayesian Criterium (SBC).

## 3. Results

### 3.1. Mean Plot Data by Thinnings Treatments

The evolution of the laurel growth variables (N, $D_0$, $H_0$, $D_g$, $H_g$, G, and V) in the agroforestry system before and after thinning as well as for trees removed in the thinned plots is summarized in Table 1. It should be noted that no dead trees were found throughout the experiment. The average stand volume was 125.26 m$^3$ ha$^{-1}$ ($\pm$15.39) at the age of nine years for the thinned plots. This means a mean annual increment (MAI) of 13.92 m$^3$ ha$^{-1}$ with a mean tree density of 418 trees ha$^{-1}$ ($\pm$15.73) in the traditional agroforestry system in the Sumaco Protected Forest.

**Table 1.** Evolution of the main stand variables before and after thinning in the laurel agroforestry system, Sumaco Protected Forest.

| | Stand before Thinning | | | | | | | Tree Removed | | | | | | | Stand after Thinning | | | | | | |
|---|---|---|---|---|---|---|---|---|---|---|---|---|---|---|---|---|---|---|---|---|---|
| Tr | N | $D_o$ | $H_o$ | $D_g$ | $H_g$ | G | V | N | $D_o$ | $H_o$ | $D_g$ | $H_g$ | G | V | N | $D_o$ | $H_o$ | $D_g$ | $H_g$ | G | V |
| $A_1$ | 450 | 24.0 | 18.4 | 17.3 | 14.4 | 11.2 | 114.6 | 275 | 24.0 | 18.4 | 16.8 | 14.0 | 6.6 | 64.4 | 175 | 22.4 | 18.2 | 18.1 | 15.1 | 4.6 | 50.2 |
| $A_2$ | 450 | 22.7 | 19.5 | 15.5 | 12.6 | 9.0 | 87.0 | 225 | 14.2 | 12.0 | 14.2 | 12.0 | 4.3 | 34.1 | 225 | 22.7 | 19.5 | 16.7 | 13.2 | 4.7 | 52.9 |
| $A_3$ | 425 | 24.3 | 18.5 | 15.5 | 10.9 | 10.0 | 81.5 | 225 | 17.0 | 13.0 | 11.9 | 9.1 | 3.8 | 20.7 | 200 | 24.3 | 18.5 | 19.5 | 12.8 | 6.2 | 60.8 |
| $B_1$ | 500 | 23.2 | 18.5 | 16.0 | 13.8 | 11.3 | 158.5 | 250 | 19.5 | 18.0 | 15.0 | 13.5 | 5.3 | 67.9 | 250 | 23.2 | 18.5 | 17.0 | 14.1 | 6.0 | 90.6 |
| $B_2$ | 450 | 26.7 | 21.0 | 20.6 | 16.9 | 12.3 | 196.1 | 225 | 17.7 | 20.1 | 15.1 | 15.0 | 4.4 | 44.0 | 225 | 26.7 | 21.0 | 21.0 | 16.5 | 7.9 | 152.1 |
| $B_3$ | 450 | 24.8 | 18.0 | 18.4 | 14.5 | 18.0 | 134.3 | 250 | 24.8 | 18.0 | 19.1 | 14.5 | 7.3 | 81.1 | 200 | 23.7 | 17.5 | 18.1 | 14.4 | 5.3 | 53.2 |
| $C_1$ | 450 | 26.4 | 20.0 | 18.6 | 16.2 | 13.0 | 156.8 | 150 | 26.4 | 20.0 | 15.8 | 15.0 | 3.1 | 34.7 | 300 | 26.4 | 20.0 | 20.0 | 16.8 | 9.9 | 122.1 |
| $C_2$ | 425 | 28.5 | 22.0 | 19.5 | 16.0 | 13.4 | 231.8 | 150 | 28.5 | 22.0 | 21.1 | 16.2 | 5.5 | 98.0 | 275 | 25.8 | 20.1 | 18.7 | 15.9 | 7.9 | 133.8 |
| $C_3$ | 400 | 29.4 | 19.0 | 19.0 | 13.8 | 12.2 | 129.3 | 150 | 25.0 | 15.5 | 17.2 | 12.0 | 3.7 | 34.6 | 250 | 29.4 | 19.0 | 20.1 | 14.9 | 8.5 | 94.7 |
| $D_1$ | 375 | 25.9 | 18.9 | 16.4 | 12.8 | 7.6 | 84.2 | | | | | | | | | | | | | | |
| $D_2$ | 350 | 21.1 | 17.8 | 15.7 | 12.5 | 6.5 | 73.7 | | | | | | | | | | | | | | |
| $D_3$ | 300 | 21.2 | 19 | 12.7 | 10.4 | 3.7 | 55.5 | | | | | | | | | | | | | | |

N = number of tree ha$^{-1}$; $D_o$= maximum diameter in plot; $H_o$ = maximum height in plot; $D_g$ = DBH average; $H_g$ = Height average; G = basal area in m$^2$ ha$^{-1}$; V = total volume in m$^3$ ha$^{-1}$.

### 3.2. Quantitative and Qualitative of Thinnings Levels

A quantitative and qualitative description of thinning levels is shown by determining the thinning weights applied to each plot (Table 2). The number of trees removed in the intensive thinning (A plots) was 54.68% on average, 48.19% of basal area, and 45.31% of volume. In contrast, the average thinning level in the light thinning (C plots) was 35.37% of trees removed, 31.81% of basal area, and 30.85% of volume. The qualitative weight index of the thinning (NG) was relatively uniformly distributed among thinning levels, with 1.16 on average, varying from 0.86 to 1.40 across thinned plots.

**Table 2.** Quantitative and qualitative weights of the thinning levels applied to laurel in the Traditional Agroforestry System, Sumaco Protected Forest.

| Plot | Quantitative Weights (%) | | | Qualitative Weight |
|---|---|---|---|---|
| | TN | TG | TV | NG |
| A$_1$ | 61.11 | 58.93 | 57.46 | 1.04 |
| A$_2$ | 50.00 | 47.83 | 45.69 | 1.05 |
| A$_3$ | 52.94 | 37.80 | 32.77 | 1.40 |
| Average: | 54.68 | 48.19 | 45.31 | 1.16 |
| B$_1$ | 50.00 | 46.90 | 46.98 | 1.07 |
| B$_2$ | 50.00 | 35.77 | 32.74 | 1.40 |
| B$_3$ | 55.56 | 58.02 | 57.92 | 0.96 |
| Average: | 51.85 | 46.90 | 39.86 | 1.14 |
| C$_1$ | 33.33 | 23.92 | 25.35 | 1.39 |
| C$_2$ | 35.29 | 41.19 | 39.30 | 0.86 |
| C$_3$ | 37.50 | 30.33 | 27.91 | 1.24 |
| Average: | 35.37 | 31.81 | 30.85 | 1.16 |

Note: TN = tree number, TG = basal area, TV = volumen, NG = relationship between number of trees and basal area.

### 3.3. Effects of Thinning Levels on Tree Parameters

A significant effect *p* (<0.0001) of thinning level was found on the absolute and relative growth rate in diameter and basal area of laurel trees (Table S1). However, the variance analysis also showed that, at the beginning of the study, there were already differences in the DBH of standing trees between the thinned and control plots. Because of this, part of the differences in absolute DBH and basal area increase between thinned and control plots are due to the initial differences in tree size. Trees in the plots with the intensive thinning (A plots) grew an average of 2.03 cm in DBH and 61.37 cm$^2$ of basal area in a year, more than double and triple, respectively, than trees in the control plots (D plots), which grew only 0.93 cm in diameter and 20.95 cm$^2$ in basal area (Table 3). Trees in the plots with moderate and light thinning levels also showed greater absolute growth in DBH and basal area as compared to control plots, but the differences were much smaller and were mainly due to differences in initial DBH of trees since, when relative growth rates were compared, there were no differences from the control (Table 3). After considering the initial tree size of the standing trees left from thinning, only trees in plots with intensive thinning (A plots) showed a higher growth rate in DBH and basal area, 58 to 60% higher than in the other plots.

**Table 3.** Average values of initial DBH (D2016) and of absolute (AGR) and relative growth rate (RGR) in DBH and basal area of laurel tree in plots subjected to different thinning levels, Sumaco Protected Forest.

| Thinning Weights | D$_{2016}$ (cm) | AGR (cm year$^{-1}$) | | RGR (cm cm$^{-1}$ year$^{-1}$) | |
|---|---|---|---|---|---|
| | | DBH (cm) | Basal Area (cm$^2$) | DBH (cm) | Basal Area (cm$^2$) |
| A | 18.12 [a] | 2.03 [a] | 61.37 [a] | 0.111 [a] | 0.223 [a] |
| B | 18.71 [a] | 1.34 [b] | 40.26 [b] | 0.072 [b] | 0.143 [b] |
| C | 19.58 [a] | 1.25 [b] | 39.45 [b] | 0.066 [b] | 0.131 [b] |
| D (Control) | 14.28 [b] | 0.93 [c] | 20.95 [c] | 0.070 [b] | 0.139 [b] |

Average values in a column followed by different letter are statistically different from each other (*p* < 0.05).

### 3.4. Relationship between Growth Rate and Plot Density in Terms of N and G

Absolute growth in DBH and basal area was significantly correlated with the number of trees per hectare, N (Figure 2A), whereas relative growth in DBH and basal area was significantly correlated with G (Figure 2B). Linear regression equations had similar fit

and were more efficient than quadratic equations to estimate growth rate in DBH and basal area, both in absolute (Table S2) and relative (Table S3) terms, from number of trees (N), respectively. Standard errors of parameters estimated in the linear model were much smaller.

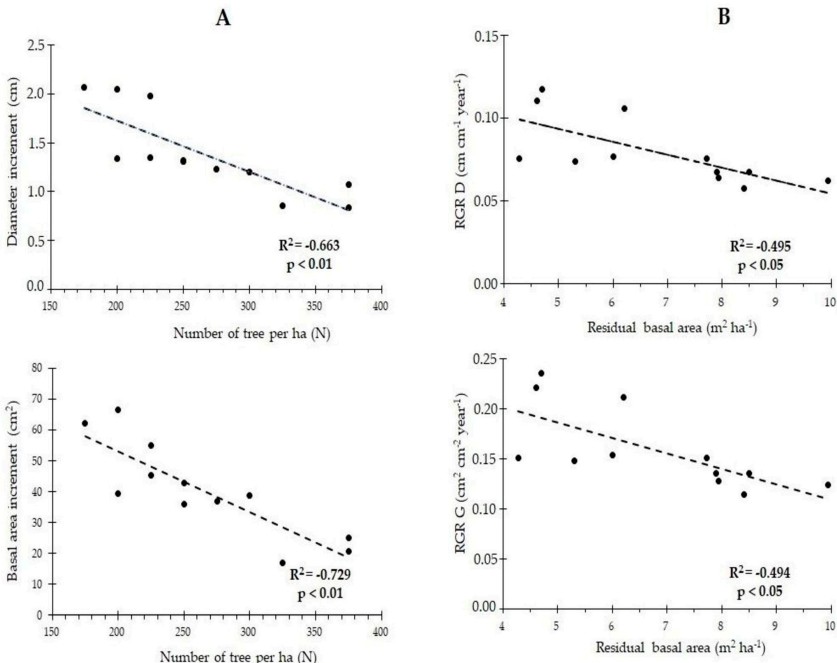

**Figure 2.** Coefficient of determination ($R^2$) and *p* value for the relationship of absolute (**A**) and relative (**B**) growth rate in DBH and basal area with plot density in terms of number of trees (N) and basal area (G) per ha, in traditional agroforestry systems in Ecuadorian Amazon upper basin.

*3.5. Growth after Thinning Treatments*

Figure 3 shows the volume evolution of laurel trees in thinning plots for the 2006–2017 period. High statistical significance ($p < 0.001$) was detected for volume (m$^3$ ha$^{-1}$) in 2017 after treatments. Tukey's 5% multiple comparison test classified the means into two ranges. The first was for intensive thinning with a mean increase of 14.3 m$^3$ ha$^{-1}$.

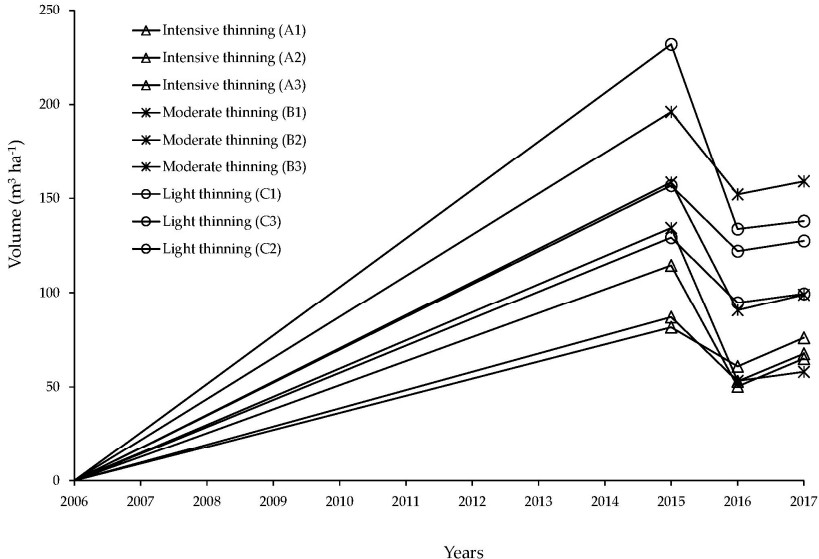

**Figure 3.** Volume evolution of laurel tree in thinning plots for the period 2006–2017, in traditional agroforestry systems in the Ecuadorian Amazon upper basin.

### 3.6. Height Diameter Relation

The trend of $H_o/D_o$ ratios per diameter class in residual trees after applying the thinning treatments shows the large heterogeneity of tree structure in these stands generated from natural regeneration (Figure 4). Trees with a high $H_o/D_o$ ratio in all diameter classes were eliminated with the intensive thinning.

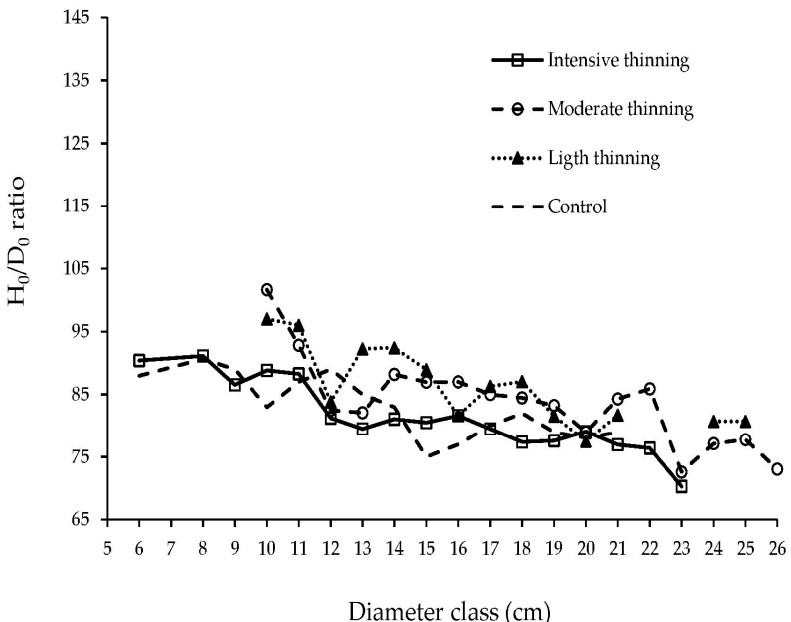

**Figure 4.** $H_o/D_o$ ratio by diameter classes and thinning treatments in laurel stands, Ecuadorian Amazon upper basin.

## 4. Discussion

### 4.1. Overall Production Observen in Thinning Plots

A MAI of 13.9 m³ ha⁻¹ year⁻¹ in the nine-year-old laurel stand was calculated for the traditional agroforestry system in the research plots. Alder and Montenegro [28] estimated a maximum MAI of 30 m³ ha⁻¹ year⁻¹ for the best sites and, for the average sites, 16 m³ ha⁻¹ year⁻¹ in rich soils of the Ecuadorian Lowland Coast. This difference in laurel growth reflects the different agro-ecological conditions existing on the Ecuadorian Coast and the Amazon region. Heuveldop et al. [29] reported a growth rate of 14.3 m³ ha⁻¹ year⁻¹ in a seven-year-old laurel agroforestry system with 278 trees ha⁻¹ at 600 m.a.s.l. in Costa Rica, and Somarriba and Beer [30] registered a MAI in volume of 20.3 m³ ha⁻¹ year⁻¹ in various agroforestry systems in the same country.

A total yield of 125.26 m³ ha⁻¹ (±15.39) is estimated at nine years of age, considering an average density of 418 trees ha⁻¹ (±15.73) for the research plots of this traditional laurel agroforestry system. In the Coca low region of the Orellana province, Peck and Bishop [31] calculated 200 m³ ha⁻¹ of laurel commercial volume with a rotation of 15 to 20 years in agroforestry systems, with data for four years of measurements in 25 permanent growth plots in agroforestry systems. Cañadas-López et al. [32] projected a laurel total volume yield of 359 m³ ha⁻¹ with an 18-year rotation on the best sites and 93 m³ ha⁻¹ with a 30-year rotation at poor sites in the Amazon upper basin. Somarriba et al. [21] estimated a volume yield of 47.7 m³ ha⁻¹ at 9 years of age with 204 trees ha⁻¹ for laurel in Central America.

Growing trees outside the forest is a feasible economic alternative for many Amazon smallholders. Therefore, the potential of local low-input tree growth is often undervalued [6,33,34]. Hoch et al. [6] mentioned that successful Amazon reforestation needs to capture local knowledge and experiences on smallholder tree growing. Therefore, it is necessary to carry out additional studies on the economic viability of this local traditional agroforestry system.

### 4.2. Quantitative and Qualitative Thinning Weights

Quantitatively, the intensive thinning treatment reduced the number of trees by 54.7%, leaving a total of 233 trees ha$^{-1}$. This number of trees is above that reported by Somarriba [18] with an average density of 167 (maximum 290, minimum 68) trees ha$^{-1}$ in Siquirres-Cahuita-Puerto Viejo, Costa Rica. Basal area in the intensively thinned plots was reduced from 10.1 m$^2$ ha$^{-1}$ to 5.20 m$^2$ ha$^{-1}$ (48.19%). Based on evidence from the traditional agroforestry system in Limón, Costa Rica, Beer et al. [35] and Somarriba and Beer [30] established a minimum basal area of 10 m$^2$ ha$^{-1}$ when laurel is associated with *C. arabica* L. (coffee) and *T. cacao* L. (cocoa) production.

The intensive thinning plots had a standing volume of 21.4 m$^3$ ha$^{-1}$ at nine years of age, which was reduced by 45.3%. Kapp and Beer [36] found a total stem volume of 96 m$^3$ ha$^{-1}$ at five years of age in agro-silvicultural plots with a productivity of 19.2 m$^3$ ha$^{-1}$ year$^{-1}$ in Costa Rican Atlantic lowlands and highlighted that these values were higher than in pure laurel stands (12 m$^3$ ha$^{-1}$ year$^{-1}$) for high-quality sites. In addition, thinning was done at nine years in the traditional agroforestry system in the Amazon upper basin, while Kapp and Beer [36] thinned at five years in the agroforestry systems in Costa Rica. Somarriba et al. [37] suggested that natural regeneration of laurel should be thinned early, within the first 10 years, leaving between 175 and 225 trees ha$^{-1}$ to establish agroforestry crops. Cañadas-López et al. [32] found a high growth rate of laurel during the first 4–5 years at the Amazon upper basin, so to take advantage of this early growth rate in laurel natural regeneration, a total of 200 well-spaced laurel trees should be left after agricultural or pasture use.

The NG average value used as qualitative thinning index was 1.15, above the unity. Consequently, all plots in this traditional agroforestry system have been thinned from below according to Abetz [38] and Kramer [39].

### 4.3. Effects of Thinning Level on Tree Growth

Statistical differences in both absolute and relative growth rate in DBH and basal area were found between thinning levels of the traditional agroforestry system in the Amazon upper basin. Research plots showed differences in individual tree size of laurel. The use of relative growth rate is a standardized growth measure for growth analysis of trees differing in initial size and has been established as a useful tool for analysis of growth performance and efficiency of tree individuals and populations [40,41]. This procedure was used in the present study.

Kramer and Röös [42] applied a regression analysis to find the relationship between the initial diameter of each tree and its growth after thinning. They concluded that the higher the diameter class, the smaller the difference between thinning treatments. The average maximum DBH in the thinned plots was 26.7 cm (Table 1). According to the quantitative thinning weights and NG values, light thinning favored growth of intermediate trees [43,44]. Hence, the observed differences between the moderate and light thinning treatments were much smaller compared to the control plots without thinning.

An increase of 2.03 cm year$^{-1}$ was obtained for the intensive thinning, with an average density of 200 trees ha$^{-1}$. Laurel, like other fast-growing species, grows better when it has ample space, and its crown can extend, especially at early ages when the highest height growth rate is achieved [17]. This pattern was confirmed by the results of this investigation, where DBH and basal area increase were evident one year after thinning, particularly in the intensive thinning (A plots). These data illustrate the sensitivity of laurel to stand density; similar results were found by Alder and Monatenegro [28]. DBH increase in the A plots in the Amazon upper basin can be compared to the growth of 2.9 cm year$^{-1}$ with a density of 60 trees ha$^{-1}$ in the province of Limón, Costa Rica and the 1.8 cm year$^{-1}$ growth with 260 trees ha$^{-1}$ and a coffee plantation under the laurel canopy in the same country [30]. According to Vega [44], laurel showed a diameter increase of 2.1–3.5 cm year$^{-1}$ in good site index locations in Suriname. Piotto et al. [45] observed a growth of 2.09 cm year$^{-1}$ at ages 5–10 years in pure laurel stand plantations in the lower Atlantic part of Costa Rica.

In contrast, control (D) plots showed a DBH growth of 0.93 cm year$^{-1}$ with an average density of 375 trees ha$^{-1}$ at the Amazon upper basin. Similar diameter increases (0.9 cm year$^{-1}$) were registered at Limón province, Costa Rica, in dense agroforestry systems (290 trees ha$^{-1}$) [30]. In poor laurel site indexes in Suriname DBH, growth varied from 0.6 cm year$^{-1}$ to 1.1 cm year$^{-1}$ [44].

*4.4. Linear Relation of Growth Rate in DBH and Basal Area with Stand Density in Terms of N and G*

The inversely proportional relationship obtained for DBH and basal area increase with density in these laurel plots fits within the self-thinning law, where diameter growth decreases with increasing tree density [46]. Plots shown in Figure 2A are similar to density diagrams that mark three points of interest in the management of traditional agroforestry systems. They allow determining the maximum limits that a laurel tree population can reach with a certain tree size. They are useful to assess whether agroforestry systems are deficient in relation to their appropriate tree density and when tree density in the agroforestry systems is such that mortality begins due to inter-tree competition.

There was also an inversely proportional relationship of relative growth rate (RGR) of D and G with basal area of the agroforestry traditional system (Figure 2B). In tropical forests, when a tree reaches the forest canopy gap, it will capture more light, producing an irregular forest structure. The stand canopy structure heterogeneity causes a three-dimension variation in light and, therefore, a differentiation of stem sizes. This causes asymmetric competition, where large trees have an advantage over small ones [47]. Weiner [48] emphasized that, if trees vary in RGR, asymmetric competition will act to increase RGR and, therefore, exaggerate differences in relative sizes. Additionally, the effect of one-side competition does not influence reproductive cycles, while growth by size and mortality are regulated by one-side competition and promote the stable coexistence of tropical trees [49].

Hummel [50] discussed the key stages of laurel stand development in Costa Rica. He proposed that competition starts at 15% and the lower limit of self-thinning occurs at 50%–60% of relative density, based on basal area. The basal area estimated for laurel was 3.39 m$^2$ ha$^{-1}$, which corresponds to the starting point of competition. Dawkins [51] predicted a basal area of 3.27 m$^2$ ha$^{-1}$ for *C. alliodora* at crown closure. The basal area average of the intensive thinning was 5.1 m$^2$ ha$^{-1}$, suggesting that the thinning practiced in traditional agroforestry systems in the Ecuadorian Amazon upper basin was out of time.

*4.5. Growth and Tree Density*

The less dense stands showed higher growth one year after thinning. This could be explained by the relationship found in the nutrient-poor soils of the Amazon region. Thus, Grau et al. [52] emphasized that sites with a certain N and P total content in the soil and a low N:P ratio would allow a faster decomposition of litter and nutrient release should be higher. This condition supports less dense stands and larger quadratic diameters. While sites with low levels of total N and P and a high N:P ratio showed denser stands and low quadratic diameters.

*4.6. Height Diameter Relation*

An important factor to consider for agro-silvopastoral systems at the Amazon region is the frequency of strong descending wind events called downburst, which are one of the major causes of tree mortality in the Amazon. Downburst effects could be seen on land areas from 30 × 30 m up to hundreds of hectares in the Amazon region [53]. Length of the living crown together with H/D ratio are considered the best parameters to describe tree stability [54], and this relationship is shaped by stand density. For this reason, trees growing in dense stands tend to be slender [55]. In a case study on planted agroforestry systems conducted in France with different species, appropriate H/D values at age eight years were below 100. Fast-growing species in agroforestry systems showed lower H/D ratio than in forest plantations used as controls [56]. Intensive thinning reduced the number of trees with

low H/D ratio, an operation that improves tree stability of traditional agroforestry systems, conferring resistance to the downburst winds at the Ecuadorian Amazon upper basin.

## 5. Conclusions

According to the high growth rates reported for laurel trees during the first 4–5 years in traditional agroforestry systems and to the results obtained in the present study, natural regeneration trees of laurel should be selected and thinned at early ages, leaving about 200 trees ha$^{-1}$ to establish future traditional agroforestry systems. Because of the reduced competition, fast growth, lower tree H/D relations, and higher quality of remaining trees will be expected.

Management of natural regeneration of laurel is a widespread practice among smallholders at the Amazon upper basin. It is an existing forest resource that requires very few inputs, offering environmental benefits, producing wood, and conserving the biodiversity. Thus, density management is a key variable for understanding the interaction between forest structure and dynamics in agroforestry systems. It should be noted that this research provides preliminary results. Reliable outcomes can be obtained only by long-term observations based on much richer research material.

**Supplementary Materials:** The following supporting information can be downloaded at: https://www.mdpi.com/article/10.3390/f14040667/s1, Table S1: Variance analysis results (probability value, P) of the initial DBH (D$_{2016}$) and the absolute (AGR) and relative growth rate (RGR) in DBH and basal area of laurel trees in plots subjected to different thinning levels, Sumaco Protected Forest; Table S2: Parameters estimated and efficiency indicators for linear and quadratic regression models fitted for absolute growth in DBH and basal area (G) on number of trees per ha (N); Table S3: Parameters estimated and efficiency indicators for linear and quadratic regression models fitted for relative growth in DBH (RGRD) and basal area (RGRG) on basal area per ha (G).

**Author Contributions:** Á.C.-L. carried out the field data compilation, Á.C.-L., P.G.-T., S.B.-G. and B.M.-T. analyzed, designed the Tables-Figures, and wrote the drafts. Á.C.-L. and J.J.V.-H. reviewed drafts of the paper and improved the statistical analysis. C.W. reviewed drafts of the paper. All authors have read and agreed to the published version of the manuscript.

**Funding:** This research received no external funding.

**Data Availability Statement:** Not applicable.

**Acknowledgments:** We thank the General Director of the National Institute of Agricultural Research (INIAP) for providing the necessary logistics to carry out this project and Jorge Elis, of the Natural Sciences Sector Program of UNESCO for the collaboration provided under the United Nations Yasuní Program and and the Federation of Indigenous Organisations of Napo (FOIN) on behalf of Bertila Avilés for the coordination with communities and data collection. We thank, too, the Faculty of Biology of the Central University of Ecuador, Gorky Gómez Díaz for providing the funds for the publication.

**Conflicts of Interest:** The authors declare no conflict of interest.

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
