# Peer review of "Thinning Levels of Laurel Natural Regeneration to Establish Traditional Agroforestry Systems, Ecuadorian Amazon Upper Basin"

_forests, doi:10.3390/f14040667_

Round 1

Reviewer 1 Report

The manuscript needs corrections and explanations. 

Reviewer 2 Report

Describing different silvicultural approach to tree intensity thinning provide data that can be used to better manage laurel stem growth to improve agroforestry systems. In my opinion, therefore, the topic of the work is accurately chosen and worthy of publication. It is also in this context that the objectives of the work, which are not stated anywhere in literal terms, can be clearly specified. 

As to the content of the reviewed manuscript, my comments is arranged in following point:

1.     At the end of the introduction, please state the aims of the work

2.     Lines 121-128, In the description of the methodology it is unclear to me how the 12 study plots covering a total of 0.48 ha in a large area covered 1.0 km x 0.5 km were determined. This is important and the results depend on it therefore it should be described more precisely. 

3.     Lines 151-156 - I propose to provide a linear model of analysis in place of the confusing description. What does subsampling mean, were not all trees analysed?

4.     Table 3. In my opinion, Table 3 could be moved to the supplement without detriment to the understanding of the results. The Anova result and the grouping of the homogenous group are shown in Table 4. 

5.     Figure 2: The caption should be changed. Person's corr refers to a linear relationship. In the case of linear regression, it is worth stating the probability of the significance of the model. 

6.     Lines 229 -233. The description of the nature of the relationship between the analysed parameters introduces confusion. It is not clear why the authors present the same relationship comparing linear regression and quadratic equations. Please choose one of the methods that better describes the relationship. If the same data are involved, the model parameters AIC and BIC unambiguously indicate a more fitting model. This does not bring about a revolution in the interpretation of the results because, as the authors note, the R2 parameters determining the strength of the relationship are similar.   I propose to present the correlations in figure 2 and move tables 5 and 6 to the supplement. 

7.     Figure 3: Why are the ratio h/dbh was not presented for the controls? 

8.     The discussion is extensive but does not add much to the interpretation of the results. In my opinion, it can be slightly reduced without detriment. 

9.     Line 326 cited is a paper [24] removed from the literature list. Please correct this. 

10.  Line 423 in the conclusion the authors refer to economic evaluation on real data. In my opinion, the study did not address this. Only incremental parameters were compared, and no economic analyses were conducted. This conclusion is wishful thinking and does not derive from the research presented.

In my opinion, the text needs some corrections but the subject it is interesting for the readers.

Reviewer 3 Report

Natural forest regeneration is very common in modern forestry, but less so in agro-forestry systems. For economic reasons, the potential value of secondary forest regeneration is not always appreciated on fallow and degraded land. Meanwhile, ecological reclamation of agricultural land using natural regeneration can be both an effective method of reforestation and an effective way to protect biodiversity.

The scientific work presented for review therefore concerns an important practical issue of natural regeneration of forest communities with Cordia alliodora (Ruiz&Pav.) Oken (laurel) on post-agricultural land, as well as the unification of silvicultural techniques in agro-forestry systems.

An experiment to assess the effect of different tree densities on the stem growth of Cordia alliodora (Ruiz&Pav.) has not been well described. In the methodological part, the authors did not fully explain the methodological solutions used. They didn't write:

· on the basis of which research the size of square field plots of 400 m2?

· what determined the pattern of distribution of square plots with different levels of thinning (this applies to both spatial distribution and location in relation to cardinal directions)

· was the boundary effect analyzed on plots with different intensity of thinning? How far apart were the plots from each other?

·  why a different number of plots was used?: control (2) and moderate (4) (Figure 1)

· why field plots in the "Control" (D) variant were designated only in fragments that were characterized by a small number of trees?

The main results focused on the description of the effects of the thinnings carried out. However, there is no information in Chapter 3 "Results" on how the forest parameters developed after thinning (in 2016)? Failure to include this information does not allow for proper interpretation of the results obtained. It is therefore necessary to add a table with tree parameters (DBH and height) before thinning in 2015 and after thinning in 2016 (mean value, standard deviation or coefficient of variation). Moreover, the results of Do (maximum diameter in plot) and Ho (maximum height in plot) presented in Table 1 are not good indicators for assessing the effects of thinning. The work also does not say whether dead trees appeared in 2016

The obtained results, due to the short period of research, do not give grounds for a general interpretation. Therefore, it should be emphasized in the paper that these are only preliminary research results, because the analyzed factors are characterized by periodic variability of growth conditions (e.g. due to weather conditions). Reliable results are provided only by long-term observations based on much richer research material.

Detailed notes on parts of the text :

Line 129-130 - It should be written: " Four three thinning levels were applied". In control (D) no thinning was done .

Line 183-185 - These results are not included in Table 1. The table should be supplemented with these results, and the chapter "Material and Methods" should contain information that such indicators have been calculated.

Line 264-265 - explain what the indices (a, b, c) placed next to the numerical values mean

Line 304-307 - no diagram in Fig. 3: H/d ratio by diameter classes in the control variant (D)

Round 2

Reviewer 1 Report

The manuscript is improved. There are a lot of data changes.